# A New Zebrafish Model to Measure Neuronal α-Synuclein Clearance In Vivo

**DOI:** 10.3390/genes13050868

**Published:** 2022-05-12

**Authors:** Ana Lopez, Alena Gorb, Nuno Palha, Angeleen Fleming, David C. Rubinsztein

**Affiliations:** 1Cambridge Institute for Medical Research, Cambridge Biomedical Campus, The Keith Peters Building Cambridge, Hills Road, Cambridge CB2 0XY, UK; al653@cam.ac.uk; 2UK Dementia Research Institute, Cambridge Biomedical Campus, The Keith Peters Building Cambridge, Hills Road, Cambridge CB2 0XY, UK; 3Department of Physiology, Development and Neuroscience, University of Cambridge, Downing Street, Cambridge CB2 3DY, UK; alenagorb02@gmail.com; 4Neuroscience and Immunoinflammation Therapeutic Area, Institut de Recherche Servier, 125 Chemin de Ronde, 78290 Croissy-sur-Seine, France; nuno.ribeiro-palha@servier.com

**Keywords:** α-synuclein, Parkinson’s disease, zebrafish disease model, neurodegeneration, autophagy, protein clearance, aggregation, axonal transport

## Abstract

The accumulation and aggregation of α-synuclein (α-SYN) is a common characteristic of synucleinopathies, such as Parkinson’s Disease (PD), Dementia with Lewy Bodies (DLB) or Multiple System Atrophy (MSA). Multiplications of the wildtype gene of α-SYN (*SNCA*) and most point mutations make α-SYN more aggregate-prone, and are associated with mitochondrial defects, trafficking obstruction, and impaired proteostasis, which contribute to elevated neuronal death. Here, we present new zebrafish models expressing either human wildtype (wt), or A53T mutant, α-SYN that recapitulate the above-mentioned hallmarks of synucleinopathies. The appropriate clearance of toxic α-SYN has been previously shown to play a key role in maintaining cell homeostasis and survival. However, the paucity of models to investigate α-SYN degradation in vivo limits our understanding of this process. Based on our recently described imaging method for measuring tau protein clearance in neurons in living zebrafish, we fused human *SNCA* to the photoconvertible protein Dendra2 which enabled analyses of wt and A53T α-SYN clearance kinetics in vivo. Moreover, these zebrafish models can be used to investigate the kinetics of α-SYN aggregation and to study the mechanisms, and potential new targets, controlling the clearance of both soluble and aggregated α-SYN.

## 1. Introduction

The small soluble protein, α-Synuclein (α-SYN), is mainly localised at presynaptic neuronal terminals, and is involved in the assembly of SNARE complexes and the regulation of neurotransmitter release from synaptic vesicles [1,2]. The intracellular accumulation and aggregation of α-SYN is associated with the initiation and progression of multiple neurodegenerative diseases called synucleinopathies, including Parkinson’s disease (PD) [3]. α-SYN is the major component of Lewy bodies (LB), and is the major pathological hallmark of both sporadic and genetic forms of PD [4,5].

α-SYN is encoded by the *SNCA* gene and its mutations or multiplications are associated with increased aggregation of α-SYN in patients and animal models, causing dominant familial parkinsonism [6,7,8,9,10]. Thus, many therapeutic strategies aim to reduce α-SYN levels and LB formation to prevent further neurodegeneration [11]. Whereas most *SNCA* mutations induce α-SYN aggregation, other pathogenic forms, such as G51D and A53E, inhibit aggregate formation [12,13]. Thus, these mutations may cause cytotoxicity via mechanisms other than aggregation.

Multiple cellular protein degradation systems can clear α-SYN, such as macro-autophagy, chaperone-mediated autophagy (CMA) and the proteasome [14,15]. α-SYN can cause autophagic dysfunction at different stages of its itinerary, from defects in autophagosome formation to lysosomal fusion. Alterations in CMA and proteasomal pathways have also been reported in the brains of PD patients and correlated with α-SYN accumulation [16,17,18]. A53T was the first point mutation described in *SNCA* that causes early-onset autosomal dominant PD [19]. Multiple studies have suggested that A53T α-SYN can contribute to the impairment of all clearance pathways previously mentioned. The effects of A53T α-SYN expression on autophagy were first described in PC12 cells, and included an increase in the number of autophagic vesicles and defects in lysosomal function, compared to cells expressing wt α-SYN. In addition, A53T α-SYN decreased proteasomal chymotrypsin-like activity, correlating with the accumulation of polyubiquitinated proteins in cell lysates [20]. A53T α-SYN binds with high affinity to the lysosome-associated membrane protein LAMP2A, resulting in blockage of CMA and failure in A53T α-SYN translocation into the lysosome for degradation [14,21]. Recently, Morgan et al., have shown that the expression of A53T α-SYN leads to the binding of mutant α-SYN to LC3B monomers in neurons derived from human pluripotent stem cells (hPSC). The mutant α-SYN-LC3B complex forms aggregates that localise to the periphery of late endosomes and promote secretion of α-SYN via the exosomal pathway, compared to its wt α-SYN isogenic controls [22].

Besides the effects of A53T on protein clearance systems, mutant α-SYN leads to other pathological features, including α-SYN aggregation and mitochondrial defects [14,20,21,22,23,24,25]. The impairment of mitochondrial function and morphology have also been associated with the overexpression of wildtype *SNCA* and other α-SYN mutations in several in vivo and cell-based models [25,26,27,28,29].

The heterogeneous nature and multifactorial etiology of PD make the generation of a single PD-model, recapitulating all common neuropathological features, extremely difficult. A better understanding of α-SYN aggregation and its clearance kinetics in vivo is needed, but, to-date, there have been no suitable models to study this. The zebrafish model offers some important advantages in this regard, such as its optical transparency, vertebrate central nervous system (CNS) composition and structural organization, and straightforward genetic manipulation [30]. In this paper, we describe the generation of new zebrafish models of PD to evaluate the degradative capacity of α-SYN-wildtype (α-SYN-wt) and mutant A53T (α-SYN-A53T) by intact neurons in vivo. The fusion of the human *SNCA* gene with Dendra2 protein resulted in the expression of a single fluorescently-labelled protein (Dendra-α-SYN) amenable for imaging and for quantification of α-SYN clearance rates in living fish; a strategy we previously used for tau [31]. Furthermore, we explored which neuropathological features of PD were recapitulated in our transgenic zebrafish and found increased neuronal cell death, reduced longevity and massive protein aggregation in the axonal projections, likely accounting for the observed impairment of mitochondrial axonal transport.

This study provides a tool to investigate potential therapeutic compounds and genetic modifiers of α-SYN clearance and offers a system to investigate α-SYN aggregation and axonal transport defects.

## 2. Materials and Methods

### 2.1. Maintenance of Fish Stocks and Collection of Embryos

All zebrafish experiments were performed in accordance with the UK Animals (Scientific Procedures) Act and UK Home Office Guidelines and local Ethical Committee approvals. Studies were performed in accordance with ARRIVE guidelines. Zebrafish adults were raised on a 14 h/10 h light/dark cycle. Embryos from experimental crosses were collected and raised in embryo medium (5 mM NaCl_2_, 0.17 mM KCl, 0.33 mM CaCl_2_, 0.33 mM Mg_2_SO_4_, 5mM HEPES) at 28.5 °C in the dark and staged according to established criteria [32]. *PanN*:GAL4-VP16 driver line-driver line was a kind gift from Herwig Baier [identified as line s1101tEt in the original publication [33]. The *EIF1α*:GAL4-VP16^cu11^ line [31] was used for ubiquitous but mosaic expression of GAL4. The *ubb*:GAL4-VP16 driver fish were created in-house by injections of wildtype embryos with an ubiquitin enhancer construct driving GAL4-VP16 (*ubb*:GAL4-VP16) generated using *ubb*:ERT2-GAL4-VP16 plasmid as template, kindly provided by Sebastian S. Gerety [34]. The ERT2-GAL4-VP16 was cut out from the backbone at SalI and NheI and replaced with GAL4-VP16. The GAL4-VP16 insert was made by PCR from the same *ubb*:ERT2-GAL4-VP16 plasmid using primers to add a new restriction site for SalI at the 5′ end of GAL4-VP16 sequence (using forward primer GTTTACAGGGATCCGTCGACTCCGGAATGAAGCTACTGTCT and reverse primer TCACTATAGTTCTAGAGGCTCGA). The new transgenic line, Tg (*ubb*:GAL4-VP16, *cryaa*:RFP) [cu51Tg], has contains a red fluorescent protein reporter to aid identification of carriers and is hereafter described as *ubb*:GAL4-VP16. 

### 2.2. Generation and Microinjection of Dendra and Dendra-α-SYN Constructs

Transgenic constructs were generated via Gateway recombination using components of the zebrafish Tol2kit [35]. Dendra2 was excised from pDendra2 plasmid (Evrogen, Moscow, Russia) and subcloned into the pME-MCS vector at Nhe1 and Sac1 sites to create pME-Dendra2. PME Dendra-α-SYN constructs were synthesized by GENEWIZ^®^, using human wild-type and A53T mutant α-Synuclein from EGFP-α-synuclein-WT and EGFP-α-synuclein-A53T plasmids [36] and the Dendra sequence from pDendra2 (Evrogen, Moscow, Russia). Recombination of middle entry clones was performed according to manufacturer’s instructions from Multisite Gateway technology (ThermoFisher, Waltham, WA, USA), using p5E-UAS and p3EpolyA components of the Tol2kit within the destination vector pDestTol2CG2, which comprises EGFP driven by the cardiac myosin light chain promoter, *myl7*, in the reverse orientation as a selection marker for transgene-carriers. The sequence was flanked by two Tol2 sites targeted by the Tol2 transposase. Resulting responder constructs UAS:Dendra2-Hsa.*SNCA*, *myl7*:EGFP and UAS:Dendra2-Hsa.*SNCA*_A53T, *myl7*:EGFP are hereafter referred to as UAS:Dendra-α-SYN-wt and UAS:Dendra-α-SYN-A53T (Appendix A). pTol2_UAS:Dendra2-polyA,* myl7*:EGFP responder construct will be hereafter named UAS:Dendra.

To generate the stable lines, the responder DNA constructs encoding the transgenes UAS:Dendra, UAS:Dendra-α-SYN-wt and UAS:Dendra-α-SYN-A53T were co-injected with Tol2 transposase mRNA (at final concentrations of 100 ng/mL and 25 ng/mL respectively) in Danieau’s solution (58 mM NaCl, 0.7 mM KCl, 0.4 mM MgSO_4_·7H_2_O, 0.6 mM Ca(NO_3_)_2_, 5 mM HEPES, pH 7.6) into the cell of 1 cell-stage TL wildtype embryos. At 3 days post-fertilization (d.p.f.), embryos were selected based on the expression of the fluorescent green cardiac marker on a Leica M205 FA fluorescence microscope, then raised to adulthood. Adult G0 mosaic fish were crossed to TL wildtype fish and the F0 generation was screened to identify Dendra-α-SYN-wt and A53T founders with strong and similar expression levels by crosses to GAL4 driver lines. Selected F0 founders were then outcrossed to TL wildtype fish to establish responder transgenic lines Tg (UAS:Dendra2-Hsa.*SNCA*, *myl7*:EGFP)^cu49Tg^ and Tg (UAS:Dendra2-Hsa.*SNCA*_A53T, *myl7*:EGFP)^cu50Tg^, hereafter described as UAS:Dendra-α-SYN-wt and UAS:Dendra-α-SYN-A53T, respectively. Transgene-carriers were identified by cardiac EGFP expression driven from the *myl7*:EGFP reporter in the absence of GAL4. Transgenic line UAS:Dendra is identified in as Tg(UAS:Dendra2,*myl7*:EGFP)^cu30Tg^.

### 2.3. Experimental Crosses

Characterisation of zebrafish with pan-neuronal expression of the Dendra-α-SYN transgenes was performed using outcrosses of the responder lines UAS:Dendra-α-SYN-wt and UAS:Dendra-α-SYN-A53T to the *PanN*:GAL4-VP16 driver fish. Crosses with *ubb*:GAL4-VP16 were used for ubiquitous expression of the transgenes to assess morphological defects and for longevity analysis. The offspring from crosses of the responder fish to *EIF1α*:GAL4-VP16 driver line showed ubiquitous but mosaic expression of Dendra-α-SYN or Dendra only, which were used for visualisation of Dendra-α-SYN aggregates and clearance assays.

### 2.4. RNA Preparation and Quantitative Reverse Transcription Polymerase Chain Reaction

For analyses of embryos and larvae, 10 fish were collected from crosses of Dendra-α-SYN-WT and A53T founder fish with driver fish *PanN*:GAL4-VP16 and *ubb*:GAL4-VP16 fish at 1, 2 and 3 d.p.f. RNA was isolated using RNeasy-Plus Mini-Kit (Qiagen, Germantown, MD, USA) following manufacturer’s instructions. For adult analysis, RNA was isolated from fin-clips from adult fish tails at 16 weeks old. A total of 50 ng of RNA was then used in One-Step qPCR combining specific-primed reverse transcription to generate cDNA and real-time PCR reaction (Invitrogen, Carlsbad, CA, USA) using TaqMan Enzyme mix and customized gene-specific primers for *GAL4, Dendra* and *GAPDH* as housekeeping gene (*GAPDH* TaqMan made-to-order gene expression code 4351372 Dr03436845_g1 from Applied Biosystem; GAL4_F GCTCAAGTGCTCCAAAGAA AAACC and GAL4_R CGACACTCCCAGTTGTTCTTCA; Dendra_F ACAAGGGCATCTGCACCAT and Dendra_R AAGCGCACGTTCTGGAAGA). Triplicates from 3 independent clutches were analyzed on a StepOne Plus Real Time PCR System and StepOneTM Sofware V.2.1 (Applied Biosystems, Bedford, MA, USA, Life Technologies, Carlsbad, CA, USA). Levels of expression were represented as the relative ratio of GAL4 and Dendra, normalized to GAPDH controls by the Δ2CT method.

### 2.5. Western Blotting

Fish were culled and collected, dried from excess embryo medium, and homogenised on ice with lysis buffer containing 1% octyl-glucoside, complete protease inhibitor cocktail and PhosSTOP tablets (Sigma, Burlington, MA, USA). The tissue was homogenized by sonication and cleared by centrifugation at 7000 rpm for 5 min at 4 °C. Supernatants were diluted in 2× Laemmli Buffer at 1:1 dilution, resolved by SDS-PAGE (10% and 12% gels) and transferred to PVDF membranes. The membranes were incubated in in 5% non-fat dry milk/PBST blocking solution for 1 h at room temperature and then the relevant primary antibodies were diluted in PBST overnight at 4 °C. Membranes were washed in PBST, incubated with 1:5000 dilution of horseradish peroxidase-conjugated secondary anti-mouse or anti-rabbit antibodies (Dako, Santa Clara, CA, USA) in PBST for 1 h at room temperature and then washed again in PBST. Bands were visualised using ECL™ (GE Healthcare Bioscience, Piscataway, NJ, USA). The amount of protein was normalized to tubulin and quantified using ImageJ (Fiji) software. The following antibodies were used: mouse anti-Tubulin (1:1000; Sigma-Aldrich), rabbit anti-Dendra (1:1000; Online Antibodies), mouse anti- α-Synuclein (1:1000; Insight Biotechnology, Wembley, UK), mouse anti-pSer129 (1:500; Biolegend, San Diego, CA, USA).

### 2.6. TUNEL Assay

Terminal deoxynucleotidyl transferase dUTP nick end labelling (TUNEL) assay was performed on 10 μm cryosections using an In Situ Cell Death detection kit (TMR, Roche Diagnostics). Slides were washed in PBS followed by 0.1% TritonX-100/PBS. Sections were then permeabilized with 0.5% TritonX-100/PBST and washed again with 0.1% TritonX-100/PBS. Apoptotic nuclei were stained using In Situ Cell Death detection kit for 2 h at 37 °C, and then washed in PBS. Sections were mounted with VECTASHIELD® Hard Set containing DAPI (Vectorlabs) and imaged using GX Capture software (Version 6.2.3.0) on a GX Optical LED fluorescence microscope equipped with a GXCAM3.3 digital camera. Five transverse sections through the head of a single fish 1, 2, and 3 d.p.f. fish were initially used to identify the time-point at which α-SYN induce higher levels of cell death compared to non-expressors. Five transverse sections through the brain of 2 d.p.f. zebrafish larvae were used to quantify the number of total apoptotic neurons per fish. The mean number of TUNEL-positive neurons per fish was calculated for each genotype (*n* = 5 fish) from five sections across the brain and eyes per fish.

### 2.7. Fluorescence and Immunofluorescence Imaging

Phenotypic analyses and in vivo imaging were performed using brightfield and fluorescence imaging on a Leica M205 FA microscope equipped with an EGFP excitation filter.

To evaluate the axonal morphology upon Dendra- α-SYN expression, whole-mount antibody staining for acetylated α-tubulin was performed on 3 d.p.f. zebrafish larvae fixed in Dent’s fixative, according to standard methods [37] using the following concentrations of antibodies: 1:25 mouse anti-α-acetylated tubulin (Sigma-Aldrich) and 1:1000 secondary antibody Alexa Fluor 568 (Invitrogen). A minimum of 4 fish per group were mounted in 3% methyl cellulose for imaging and analysis on a Leica M205 FA microscope. Abnormalities in motor neuron length (innervation of ventral myotome) and bifurcation (innervation of central myotome) were quantified as previously described [31].

Immunostaining of cryosections was performed to detect axonal integrity in 3 d.p.f larvae using acetylated α-tubulin (1:50, Sigma-Aldrich) and synaptotagmin-2 (znp-1) (mouse anti-synaptotagmin 2; 1:50; ZIRC). Longitudinal sections (15 μm) across the hindbrain were rinsed in PBS and PBS 0.1% Tween-20 (PBST), then blocked in 10% goat serum/PBST and incubated with primary antibodies at 4 °C overnight. Slides were rinsed in PBST and then incubated with Alexa Fluor 568 secondary antibody (1:1000; Invitrogen) for 2 h. They were washed in PBST and mounted with VECTASHIELD^®^ Hard Set containing DAPI (Vectorlabs). A similar protocol was used to detect potential gliosis by GFAP antibody staining on transverse sections across the spinal cord of 3 d.p.f. PFA-fixed larvae (1:50 mouse anti-GFAP zfr1, ZIRC). Images were taken using a Leica SP8 laser confocal microscope at x3 magnification with a 40x objective using the 1-μm step size ‘z-stack’ tool. Images were then processed and analysed using Fiji software (ImageJ) with maximum projections of z-stacks comprising the whole neuronal projection. Images comparing two experimental groups were performed at the same time using identical settings.

Quantification of Dendra-α-SYN aggregates was performed on PFA-fixed 10 μm transverse sections through the head of 3 d.p.f. larvae and imaged using a Zeiss Axio Zoom.V16 microscope with a QImaging Retiga 2000 R digital camera. Dendra-derived fluorescent signals were preserved after fixation and aggregates were identified as bright puncta in neuronal tissue. Dendra-α-SYN puncta were manually quantified in the ventral diencephalic region from the images.

### 2.8. Behavioural Assays

Touch-response was assayed on 3 d.p.f. individual zebrafish larva with neuronal expression of Dendra-α-SYN, as previously described [31]. Three experiments from independent clutches were performed with 20 fish per genotype.

### 2.9. Longevity Assay

Offspring from crosses of responder lines to *ubb*:GAL4-VP16 were raised to adulthood and numbers were recorded periodically. In each group 90 fish were used and quantified weekly from 1 to 6 weeks, every fortnight from 6 to 12 weeks, and monthly after 12 weeks. Survival was analyzed by Log-rank (Mantle-Cox) method to compare all 4 groups and each transgenic fish with its non-expressor siblings. The length of 10 weeks-old fish with ubiquitous expression of Dendra-α-SYN-wt and A53T was measured from the head to the junction of the body/caudal fin.

### 2.10. Thioflavin-S Staining

Staining of β-sheet aggregated structures formed by Dendra- α-SYN protein was performed using a modified Thioflavin-S staining [38] on longitudinal cryosections across the heads of 3 d.p.f. larvae. Sections were immersed in potassium permanganate solution for 4 min followed by 2 min of bleach treatment. After rinsing in water, slides were incubated in 0.1% Thioflavin-S/50% ethanol for 5 min at room temperature in complete darkness. Sections were briefly washed twice in 50% ethanol solution and rinsed in water before mounting with VECTASHIELD Hard Set with DAPI. Images were taken using a GX Optical LED fluorescence microscope equipped with a GXCAM3.3 digital camera. Fish 3 d.p.f with pan-neuronal expression of Dendra-tau-A152T were used for comparison.

### 2.11. Mitochondrial Axonal Transport

Offspring from UAS:Dendra-α-SYNA53T crossed to *PanN*:GAL4-VP16 were injected with a total amount of 11.5 pg of *NeuroD*:mitoRFPTag DNA construct [39] in Danieau’s solution. Pigment formation was inhibited by the addition of 0.003% of phenylthiourea (PTU) from 24 h post-fertilization (h.p.f.) to facilitate imaging. Injected Dendra-α-SYN expressors and negative siblings were screened at 48 h.p.f. and selected for similar mosaic expression of the mitochondrial construct using RFP fluorescence filter on a Leica M205 FA microscope. At 72 h.p.f., selected fish were then anaesthetized by addition of MS222 to a final concentration of 0.133 mg/mL in embryo medium and embedded in 0.75% low melting point agarose in an imaging chamber. Images and videos of moving mitochondria along the spinal cord were performed on a Leica TCS SP8 laser confocal microscope running LAS X (version 1.8.1.13759) software, using a 40x oil-immersion lens and 2.25x zoom. Images used for the analysis of mitochondrial density and morphology were taken using a z-stack of 12 μm (1 μm step-size). Videos were taken in a single plane at 1.036 s -intervals and for a duration of 1.03 min. The quantification of mitochondrial length and numbers was performed manually using the ‘point tool’ and ‘Straight tool’ in Fiji. All mitochondria within the same-sized region of interest (ROI), localised to the center of the hyper-stack image projection of the spinal cord, were quantified. Analysis of mitochondrial motility was performed on a single plane comprising 120 μm of 1 or 2 axons in focus. All mitochondria within an image were traced using FITJI (ImageJ) software and analyzed using “detection and tracking” and “Spot detector” tools in Icy software (http://icy.bioimageanalysis.org/ (accessed on 1 October, 2020)). Mitochondria were only considered motile when the distance travelled exceeded 2 μm at a speed at least 0.1 μm/s [40]. A minimum of 4 fish were used per group.

### 2.12. Quantification of Dendra-α-SYNuclein Clearance

To determine the clearance kinetics of Dendra-α-SYN in living neurons, UAS:Dendra-α-SYN responder fish were crossed with the *EIF1α*:GAL4-VP16 driver line. The resulting mosaic expression pattern allows the visualisation, photoconversion and imaging of single neurons in the spinal cord, as previously described [41]. The photoconversion of Dendra signal from individual neurons in the spinal cord was performed by UV irradiation (405 nm) for 100 ms using the Bleachpoint tool on a confocal microscope model Leica SP8, 3x zoom with a 40x objective and LAS X (version 1.8.1.13759) software. Red signals from photoconverted neurons were imaged immediately after photoconversion and at 24 h intervals during the next 48 h. Changes in the intensity of red signals were analysed using ImageJ by selecting regions of interest ‘ROI’ around each photoconverted neuron with red Dendra-α-SYN fluorescence and measuring the ‘Integrated Density’ at different time-points. Dendra-α-SYN clearance was represented as the percentage of the initial red fluorescent intensity immediately after photoconversion, quantified at each time point.

To evaluate the clearance of Dendra-α-SYN in presence of the autophagy modulator, ammonium chloride (NH_4_Cl), fish with mosaic expression of wildtype or mutant Dendra-α-SYN-A53T were treated with 10 mM NH_4_Cl diluted in embryo medium immediately after photoconversion until 48 h.p.f. Treatment was replenished after each imaging time-point to maintain drug concentration.

### 2.13. Statistics

Statistical analysis was performed using GraphPad 8.02 software. Statistical significance was calculated using One-way ANOVA with Tukey’s multiple comparison test for multiple comparisons. Log-rank (Mantel-Cox) analysis was used for survival experiment. Comparison between 2 conditions was performed using the unpaired *t*-test. Results are presented as mean ± standard deviation (s.d.) or Standard Error of Mean (SEM). *p* < 0.05 was considered significant.

## 3. Results

### 3.1. Generation of New Transgenic Zebrafish Lines Expressing Dendra-α-SYN

Stable lines UAS:Dendra-α-SYN-wt and UAS:Dendra-α-SYN-A53T were generated from DNA-injected fish as described in the methods (Appendix A). When crossing UAS:Dendra-α-SYN lines to the *PanN*:GAL4-VP16 driver fish, offspring exhibit strong neuronal expression of the Dendra-α-SYN fusion protein, which results in a bright fluorescently labelled nervous system from 24 h.p.f. (Figure 1A). Analysis by qPCR was used to identify Dendra-α-SYN-wt and A53T founders at the F0 generation with similar expression levels, to allow comparison of the 2 transgenic lines. The offspring from selected founders crossed to *PanN*:GAL4-VP16 and *ubb*:GAL4-VP16 driver lines showed similar relative Dendra/GAPDH expression levels from 1 to 3 d.p.f. (Figure 1B). Dendra/GAPDH expression levels decreased over time and were not caused by changes in GAL4, as GAL4 expression was found to increase with larval age, a pattern that was observed in all founders analyzed (Appendix A).

As a consequence of the expression of Dendra-α-SYN transgene as a fusion protein, antibodies targeting both human α-Synuclein (14 kDa) and Dendra2 protein (26 KDa) detect the same strong band around 40 KDa from 24 h.p.f. (Appendix A) in fish with pan-neuronal expression. Their relative levels compared to a loading control are comparable for both antibodies (Appendix A) and allow quantitative analyses using either antibody. Despite the reduction in Dendra mRNA levels found from 1 to 3 d.p.f. (Figure 1B), neuronal protein levels remained high in both transgenic lines at 3 d.p.f. (Figure 1C and Appendix A). To investigate whether human α-SYN could undergo post-translational modifications (PTMs) in our zebrafish model, we examined the phosphorylation of α-SYN at residue S129, considered a disease hallmark in Parkinsonian patients [42]. The levels of phosphorylated α-SYN were significantly increased in fish expressing the A53T mutation compared to those expressing Dendra-α-SYN-wt in the CNS at the same age (Figure 1C). Although the protein band for Dendra-α-SYN was detectable from 1 d.p.f., the phosphorylation of α-Synuclein at S129 was only detectable at 3 d.p.f. (Appendix A) suggesting a time-dependent effect for this phosphorylation site. In contrast with the ubiquitous promoter *ubb*, the pan-neuronal promoter used to restrict GAL4 and Dendra-α-SYN expression to the CNS (*PanN*) does not stay on permanently, and hence, a reduction in α-Synuclein levels was observed at 6 d.p.f. (Appendix A). Due to these results, we restricted the phenotypic characterisation to ages between 1 and 3 d.p.f. when the *PanN*:GAL4-VP16 driver line was used.

Neither the pan-neuronal expression of Dendra-α-SYN-wt nor mutant A53T caused any gross morphological abnormalities, compared to non-expressor siblings after 3 days (Appendix A). Crosses to the *ubb*:GAL4-VP16 driver fish also resulted in morphologically normal larvae from 1 to 3 d.p.f. (Appendix A).

### 3.2. Neuronal Toxicity and Survival in Dendra-α-SYN Expressing Fish

To investigate if the pan-neuronal expression of Dendra-α-SYN-wt or A53T caused any toxic effect in the CNS, we measured the number of apoptotic cells by TUNEL. Initially, transverse sections across the brains of Dendra-α-SYN-wt-expressing fish and non-expressing siblings at 1, 2 and 3 d.p.f were used to determine the appropriate time-point at which cell death occurs (Appendix A). Quantification of apoptotic cells TUNEL-positive nuclei in Dendra-α-SYN-wt and A53T, compared to negative siblings, showed a 5-fold increase in the number of TUNEL-positive nuclei upon Dendra-α-SYN expression at 2 d.p.f. (Figure 2A,B). A similar level of cell death was found among the Dendra-α-SYN-expressing fish, suggesting that expression of both mutant A53T and wildtype human α-SYN can cause neuronal cytotoxicity. However, there was no evidence of gliosis in response to the elevated neuronal death (Appendix A).

We next investigated whether transgenic zebrafish showed any abnormalities in motor neuron morphology by quantifying defects in the axonal branching and pathfinding, after staining axonal tracks with acetylated tubulin in 3 d.p.f. fish (Appendix A). No significant differences were observed in Dendra-α-SYN-wt or A53T, compared to their negative siblings in any of the parameters analyzed (Appendix A). Consistent with this observation, both Dendra-α-SYN-wt and A53T expressors showed normal escape responses to tail touch stimulus (Appendix A).

The expression of the UAS:Dendra-α-SYN transgenes driven by the pan-neuronal promoter declines over time, as reflected in the decrease of Dendra-α-SYN protein levels after 6 d.p.f (Appendix A). To investigate the effects of prolonged expression of the transgene, we crossed UAS:Dendra α-SYN lines to *ubb*:GAL4-VP16 driver fish and raised the offspring to adulthood. Despite the lack of any gross morphological defects or motility impairment in zebrafish larvae with pan-neuronal α-SYN expression, adult fish with ubiquitous expression of Dendra-α-SYN showed significant differences at later stages. Juvenile zebrafish, expressing Dendra-α-SYN-A53T, were smaller in size at 10 weeks, compared to their non-expressing siblings (Figure 2C). Approximately 20% of A53T fish had clear macroscopic deformities, such as severe torsion of dorsal spine and smaller heads after 16 weeks, whereas fish with comparable mRNA levels of Dendra-α-SYN-wt remained normal (Figure 2D and Appendix A). Fish expressing Dendra-α-SYN-A53T also showed reduced lifespan with a pronounced decrease in numbers from 12 months old (from week 45), compared to transgene-negative siblings or fish expressing Dendra-α-SYN-wt (Figure 2E).

### 3.3. Dendra-α-SYN Expression Causes Aggregate Formation in the Neuronal Projections

Dendra-α-SYN zebrafish with pan-neuronal expression throughout the CNS exhibit extensive aggregate-like formations which appear as bright fluorescent puncta in the spinal cord and brain of transgenic fish (Appendix A). These puncta were more clearly visualised with mosaic transgene expression in the offspring of crosses to *EIF1α*:GAL4-VP16 driver fish and were predominantly localised in axonal projections (Figure 3A). This punctate pattern of Dendra-α-SYN along the axonal projections of the spinal cord differs from the continuous and uninterrupted Dendra signal in axons from fish expressing other neurodegenerative-associated proteins, such as mutant Dendra-tau-A152T (Appendix A). Puncta were not observed in other cell types expressing the Dendra-α-SYN transgene (Appendix A). Aggregates were similarly abundant in fish expressing Dendra-α-SYN-wt and A53T (Figure 3B and Appendix A). Confocal imaging of cryosections stained with either intracellular or membrane-bound neuronal makers, such as acetylated α-tubulin or synaptotagmin-2 (znp-1), was used to determine whether these fluorescent puncta were protein aggregates or a consequence of axonal fragmentation. Axonal morphology in the spinal cord appeared normal in Dendra-α-SYN fish (Appendix A) consistent with the absence of abnormalities in motor neuron axonal tracks previously described (Appendix A). In addition, we observed thioflavin-S-reactive material in the brains of Dendra-α-SYN-wt and mutant A53T expressors, which was not present in negative siblings or Dendra-tau-A152T mutant fish at 3 d.p.f., in agreement with the hypothesis that fluorescent puncta correspond to deposits of accumulated protein (Figure 3C).

These aggregates form very early in the developing embryo and increase in numbers over time in the brains of Dendra-α-SYN expressing fish from 24 h.p.f. to 3 d.p.f. (Appendix A). To further investigate the process of aggregate formation, we used crosses of the responder lines to the mosaic GAL4 driver, *EIF1α*:GAL4-VP16, enabling the visualisation of single axonal projections in the spinal cord. Mosaic expressors of Dendra-α-SYN-wt or A53T mutants showed aggregate formation at 22 h.p.f. (Figure 3D), often in the axonal projections and rarely seen in the cell body. A time-course analysis, from 22 h.p.f. to 4 d.p.f. using confocal imaging, revealed that aggregates can form at any time (Figure 3E) and grow in size (Figure 3F). 

### 3.4. Dendra-α-SYN Expression Causes Mitochondrial Transport Defects

Early aggregation of α-SYN in axonal projections might influence the physiological functions within this compartment, such as axonal transport. To investigate this hypothesis, we evaluated the effects of Dendra-α-SYN aggregates on mitochondrial axonal transport. The offspring from outcrosses of UAS:Dendra-α-SYN-A53T to *PanN*:GAL4-VP16 were injected with the transgenic construct *NeuroD*:mitoRFPTag, comprising a neuronal promoter driving expression of red fluorescent protein with a mitochondrial targeting sequence, that allows the visualisation of neuronal mitochondria. Injected transgene-negative siblings were used as controls for comparison. Confocal images of the spinal cord showed a great abundance of labelled mitochondria in both experimental groups (Figure 4A). Despite a comparable number of mitochondria in Dendra-α-SYNA53T and the control group (Figure 4B), mitochondrial length was significantly smaller in α-SYN expressors (Figure 4C). Large mitochondria (also brighter) were largely non-motile whereas smaller mitochondria were highly mobile (Appendix A). The abundance of labelled mitochondria and the pan-neuronal expression of Dendra-α-SYN hinder the possibility of identifying the directionality of axonal tracks and, hence, precludes discrimination between anterograde and retrograde transport of mitochondria. Nevertheless, mitochondrial transport was evaluated in all mitochondria within 126 μm segments of axons in the spinal cord for 1 min, independently of movement direction. Expression of Dendra-α-SYN impacted the proportion of moving mitochondria (Figure 4D), reducing distance travelled, duration and speed of motile mitochondria, compared to non-transgenic siblings (Figure 4E–G).

### 3.5. Analysis of the Clearance Rate of Dendra-α-SYN Protein In Vivo

The offspring with mosaic expression of Dendra-α-SYN from crosses of the responder fish with the *EIF1*α:GAL4-VP16 driver line were used to evaluate the clearance of Dendra-α-SYN protein. Using confocal microscopy, the green Dendra-derived fluorescent signal from isolated neurons in the spinal cord was photoconverted by UV light targeted to the soma of the cell. Photoconversion causes a pool of Dendra-α-SYN protein to switch from green to red fluorescence, resulting in a reduction in the green fluorescent signal of that neuron after photoconversion (Figure 5A). A time-course series of images of the red signal shows the reduction in the red fluorescence over time as a result of clearance of photoconverted Dendra-α-SYN (Figure 5B). The intensity of the red signal from the same neuron was quantified immediately after photoconversion (0 h) and at 24 h-intervals and represented as the percentage of the initial red intensity of that cell at 0 h (after photoconversion). Red Dendra-α-SYN-wt and mutant A53T signal decreased at the same rate and showed overlapping graphs (Figure 5C), indicating that Dendra-α-SYN-wt and A53T have similar in vivo clearance kinetics in these zebrafish models.

To investigate the utility of these zebrafish models for measuring perturbations of protein clearance kinetics, we performed some preliminary experiments with autophagy- lysosomal inhibition, as a proof-of-concept. Since wildtype and A53T mutant α-SYN are known to be autophagic substrates, we analyzed the contribution of this clearance pathway to the degradation of Dendra-α-SYN. Ammonium chloride, an inhibitor that blocks autophagy by compromising lysosomal acidification, reduced the clearance rate of Dendra-α-SYN-A53T (Figure 5D) and Dendra-α-SYN-wt (Appendix A), corroborating the role of autophagy in α-SYN degradation.

## 4. Discussion

The aggregation of α-SYN is one of the key pathological hallmarks in PD and other synucleinopathies, ultimately resulting in neuronal death [43]. Multiple studies support the view that the accumulation of α-SYN compromises lysosomal degradation by impairing macro-autophagy and CMA in mammalian cells and transgenic mice [14,21,24,44,45]. Moreover, α-SYN accumulation reduces proteasome catalytic activity in brain tissue from PD patients and in vivo models [46,47,48]. Nevertheless, α-SYN itself is a substrate for both lysosomal and proteasomal degradation pathways, generating a vicious cycle [49]. Hence, the control of neuronal proteostasis by these clearance systems is important in the pathogenesis of synucleinopathies, but the time-dependent interplay between the accumulation of aggregated α-SYN, clearance pathway impairment, and neuronal toxicity in vivo is still unknown.

The use of cycloheximide, an inhibitor of protein synthesis, followed by Western blotting or [35S] Methionine radio-labeled α-SYN have been previously used to determine the half-life of wildtype α-SYN, the effect of SNCA mutations, or PTMs on α-SYN, such as S129 phosphorylation or SUMOylation in cell based-experiments [50,51,52].

To investigate the in vivo clearance kinetics of α-SYN, we generated new zebrafish transgenic lines expressing human α-SYN wildtype and mutant A53T fused to Dendra2 at the N-terminal (Dendra-α-SYN). Despite the presence of other forms of synuclein such as β (β-Syn) or γ (γ-Syn), zebrafish lack an orthologue of *SNCA* [53], so interactions of Dendra-α-SYN with an endogenous α-SYN protein can be discarded. Interestingly, knockdown of β- or γ-Syn induces motor deficits in zebrafish and recapitulates some aspects observed in triple knockout mice [54,55]. In our model, Dendra-α-SYN transgenic zebrafish with pan-neuronal expression showed phosphorylation at S129 residue at 3 d.p.f., one of the most investigated and controversial PTMs in α-SYN studies. Dendra-α-SYN-A53T transgenic zebrafish showed higher levels of S129 phosphorylation than α-Syn-wt at 3 d.p.f. despite having similar levels of total and aggregated protein. Our results suggest that, despite the absence of a zebrafish α-Syn, endogenous zebrafish kinases can indeed interact with exogenous human α-SYN. The high homology between human and zebrafish kinases [56] opens up the opportunity for investigating the roles of different kinases in α-SYN phosphorylation [57], and makes this model suitable for testing drugs that target these specific kinases.

Other zebrafish models expressing α-SYN have been generated previously. Prabhudesai and colleagues used injections of HuC-α-syn-T2A-dsRed DNA to drive neuron-specific expression of human wildtype α-syn, which resulted in severe morphological abnormalities, increased neuronal apoptosis, and premature death of injected embryos [58]. O’Donnell et al., also used the UAS:GAL4 system to restrict the expression of human wild-type α-SYN to Rohon-Beard sensory neurons in order to avoid the lethality of the aforementioned model and found a moderate increased in cell death at 2–3 d.p.f. and axonal defects [59]. Different strengths of the promoters used could account for the differences between these models and our model. Ideally, for comparisons of different constructs/tags, one should try express these with the same promoter system and carefully compare fish with very similar transgene expression levels.

The main advantage of our new zebrafish models is the expression of human α-SYN fused to the green-to-red photoconvertible protein Dendra2. This fluorescent tag not only allows the visualisation of α-SYN and its potentially toxic effects in neuronal cells, as other animal models previously reported [60,61,62,63], but also allows the quantification of α-SYN clearance kinetics in living neurons. The photoconversion of green Dendra into a pool of red Dendra was used to track Dendra-α-SYN degradation rate in single neurons in the spinal cord, as previously described when investigating in vivo tau clearance kinetics in zebrafish [31].

One of the hallmarks for PD, most strikingly recapitulated in our models, is the aggregation of α-SYN. This phenomenon only occurs in neurons, especially evident in the axonal projections, whereas no aggregation was seen in other cell types with similar fluorescent signals. The aggregate formation in the axon (around 22 h.p.f.) preceded neuronal death (2 d.p.f.), a series of events also found in multiple cell-based studies. Weston et al. recently used a zebrafish model to investigate α-synuclein aggregation in vivo at presynaptic terminals. Their results suggest the existence of distinct pools of α-Syn with varying mobility determined by fluorescence recovery after photobleaching (FRAP), likely representing different populations of aggregated and non-aggregated protein, which also were independent of S129 phosphorylation [64]. The presence of similar Dendra-α-SYN aggregation in both transgenic fish and A53T mutants, but with higher levels of pS129 in A53T mutants, supports the hypothesis that S129 may not be crucial for the multimerisation of the protein.

In contrast with other zebrafish studies, where α-SYN aggregation was observed at 48 h.p.f. accompanied by axonal fragmentation [58], the early appearance of Dendra-α-SYN aggregates in our models (around 22 h.p.f) did not cause any obvious disruption of the axonal tracks or general morphological defects. Despite no changes in aggregation or cell death at earlier stages, our data shows that the expression of A53T causes morphological defects in adults and reduced lifespan, compared to fish with wildtype α-synuclein expression. Apoptosis in these adult fish may occur prior to the stages we have studied.

The aggregation of α-SYN has been previously related to microtubule disruption and defective mitochondrial axonal transport in vitro and in vivo [65]. The expression of A53T α-SYN reduced mitochondrial trafficking in mouse cortical neurons [66] and SH-SY5Y cells [29. Accordingly, we observed reduced mitochondrial motility and length in fish expressing Dendra-α-SYN-A53T, compared to non-expressor siblings. Reduced mitochondrial size could be a consequence of mitochondrial fragmentation, as suggested by other studies in cultured cells and in *Caenorhabditis elegans* [28,67]. Mahul-Mellier et al. recently demonstrated that α-SYN aggregates have high affinity binding to lipid structures, such as membranes and organelles, including mitochondria [68]. Other effects of α-SYN on axonal components, accounting for reduced mitochondrial transport, cannot be discarded. For example, alterations of the levels of motor proteins have been previously reported in PD patient’s brains, animal and cellular PD models overexpressing α-SYN [69,70,71].

In summary, this study describes in vivo models to investigate human wildtype and A53T mutant α-SYN accumulation and clearance in living zebrafish neurons. Here we report that the expression of Dendra-α-SYN induces massive protein aggregation, neuronal cell death and defects in mitochondrial axonal transport. In vivo imaging allows the potential to investigate the specific roles for autophagy and the proteasome in α-SYN degradation and how neurons respond to modulation of these clearance pathways. Moreover, the binary UAS:GAL4 system used permits the tissue-specific control of the expression of Dendra-α-SYN, allowing studies of wildtype and A53T α-SYN in different cell types i.e., neurons vs. glia.

## Figures and Tables

**Figure 1 genes-13-00868-f001:**
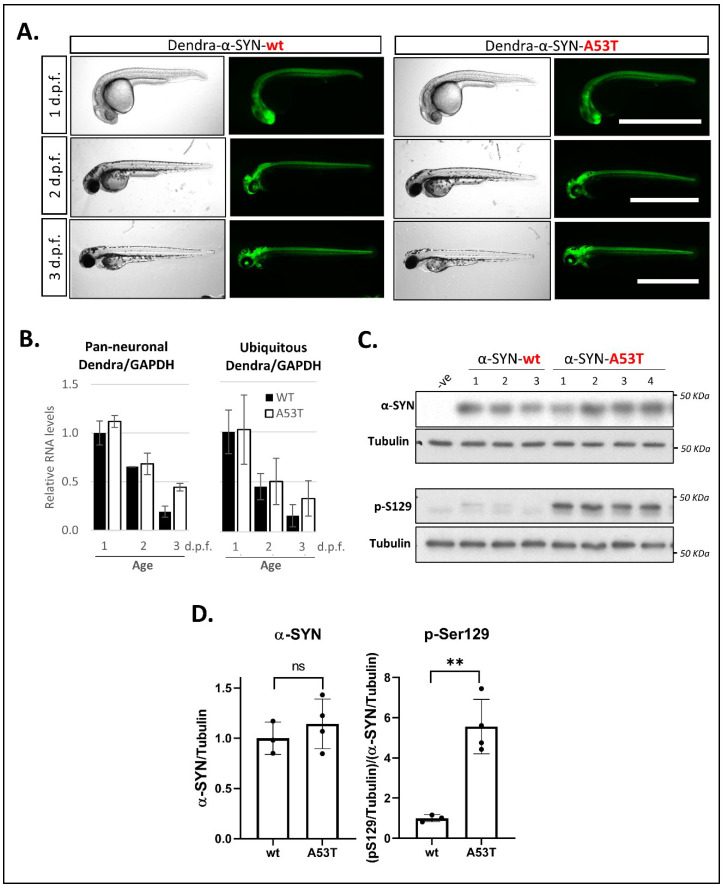
**Expression of Dendra- α-SYN in transgenic zebrafish.** (**A**) Representative brightfield and fluorescence images of Dendra-α-SYN positive offspring from stable UAS:Dendra-α-SYN-wt and A53T fish lines crossed to the PanN:Gal4VP16 driver fish at 1, 2 and 3 d.p.f. Expression localised to the CNS remains strong from 1 to 3 d.p.f. and does not result in morphological defects. Scale bar represents 1 mm. (**B**) Quantification by qPCR of the relative expression levels of Dendra/GAPDH in offspring from selected Dendra-α-SYN-wt (black) and A53T founder fish (white) showing comparable mRNA levels at 1, 2 and 3 d.p.f. when outcrossed to pan-neuronal or ubiquitous GAL4 driver lines (PanN:GAL4-VP16 and *ubb*:GAL4-VP16). Both groups showed reduced Dendra mRNA levels over time normalised by GAPDH. Data represents mean ± s.d. (*N* = 3 independent clutches for each age and genotype) (**C**) Western blot for total α-SYN and phosphorylated α-SYN at residue S129 (pS129) in whole fish lysates from 3 d.p.f. fish with pan-neuronal expression of Dendra-α-SYN-wt and A53T compared to tubulin levels as loading control. (**D**) Densitometry of total α-SYN and pS129 immunoblots in (**C**). The two transgenic lines had equivalent α-SYN levels relative to tubulin, whereas the levels of pS129 α-SYN were significantly increased in Dendra-α-SYN-A53T mutant fish, compared to α-SYN-wt (** *p* < 0.01 vs. α-SYN-wt).

**Figure 2 genes-13-00868-f002:**
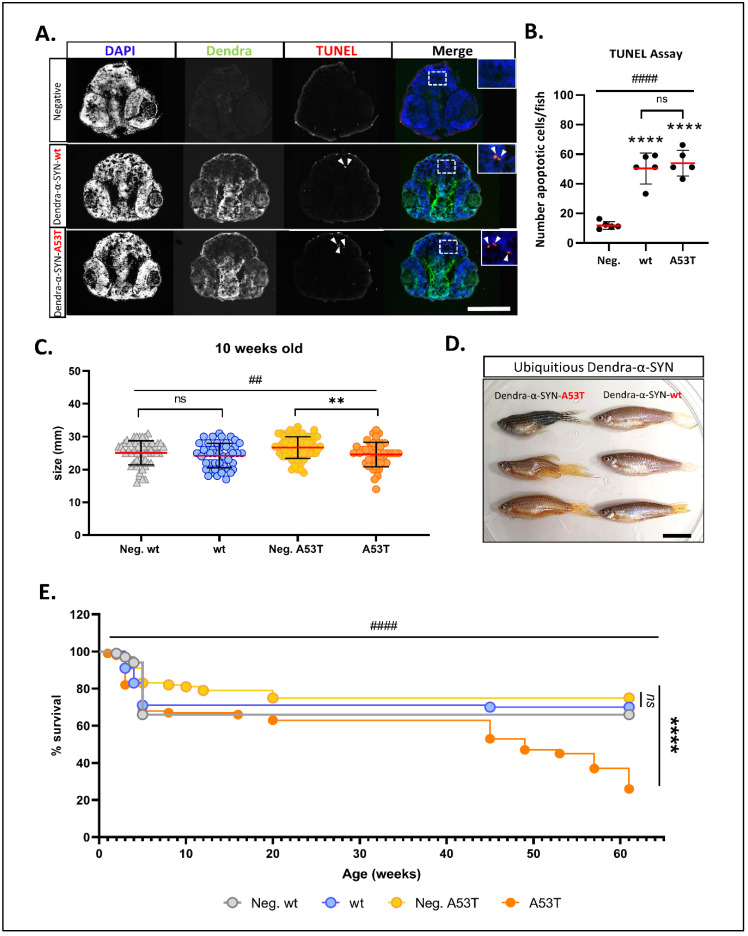
**Characterisation of neuronal pathology and longevity in Dendra-α-SYN transgenic zebrafish.** (**A**) Representative images of TUNEL labelling in the brains of larvae at 2 d.p.f. Pictures show DAPI, Dendra and TUNEL in transverse cryosections across the heads of embryos with pan-neuronal expression of Dendra-α-SYN-wt and A53T, compared to transgene-negative siblings imaged by epifluorescence microscope. Apoptotic nuclei are highlighted by white arrowheads in TUNEL images and shown at higher magnification in merge images. Scale bar represents 250 μm. (**B**) Quantification of the total number of apoptotic cells in 5 sections per fish detected by TUNEL assay. A significant increase in cell death was seen in fish expressing Dendra-α-SYN-wt (wt) or the A53T mutation, compared to transgene-negative siblings (Neg.). Data represented as mean (red line) ± s.d. (*N* = 5 fish per group, **** *p* < 0.0001 Dendra-α-SYN transgenics vs. Neg.; ns *p* > 0.05 Dendra-α-SYN-wt vs. A53T; #### *p* < 0.0001 ANOVA). (**C**) Graph represents the differences in body length of 10 weeks-old fish with ubiquitous expression of Dendra-α-SYN-wt and A53T related to their respective transgene-negative siblings (Neg. wt and Neg. A53T). Fish expressing mutant A53T showed significant reduced length (*N* = 47 fish per group, ** *p* < 0.005 vs. transgene-negative siblings; ## *p* < 0.005 ANOVA for whole data ser). (**D**) At 16 weeks old, fish with ubiquitous expression of Dendra-α-SYN-A53T (left) display overt morphological defects, such as bent spine and smaller heads, whereas Dendra-α-SYN-wt (right) appear normal. Scale bar represents 1 cm. (**E**) ubiquitous expression of Dendra-α-SYN-A53T reduced lifespan compared to Dendra-α-SYN-wt or non-expressors from 41 weeks old (*N* = 90 fish per group at beginning of the experiment, **** *p* < 0.0001 A53T vs. transgene-negative siblings; #### *p* < 0.0001 Log-rank (Mantel-Cox) for all 4 groups).

**Figure 3 genes-13-00868-f003:**
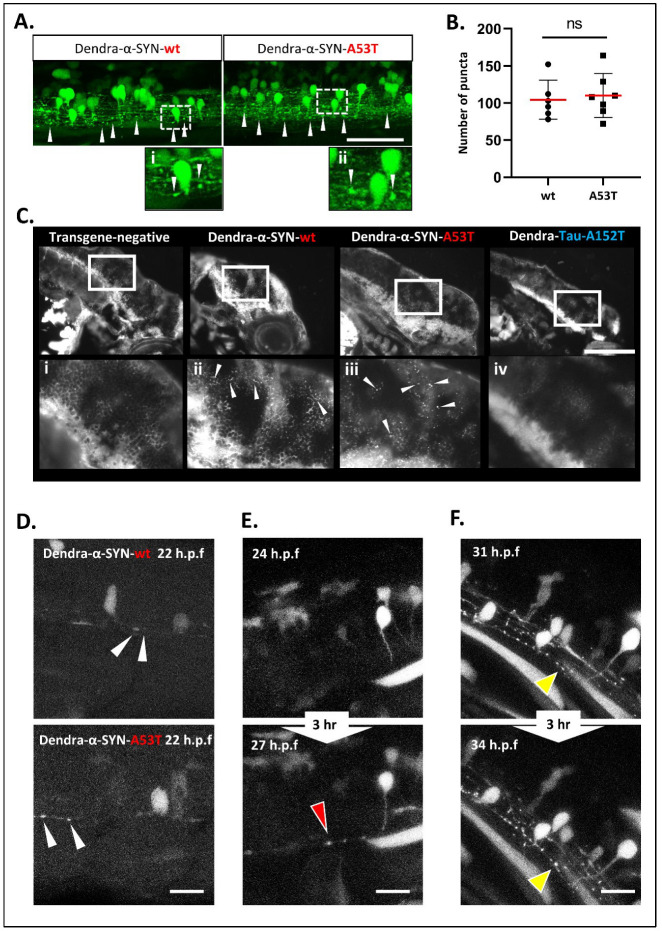
**Accumulation and aggregation of Dendra-α-SYN.** (**A**) Representative confocal images of the mosaic expression of Dendra-α-SYN in the spinal cord. Both Dendra-α-SYN-wt and A53T-expressing larvae showed bright puncta of Dendra signal along the axons (white arrowheads) in 3 d.p.f. fish. Images (i,ii) show puncta in axons in detail. Scale bar represents 150 µm. (**B**) Quantification of Dendra puncta seen in transverse sections across the head of 3 d.p.f larvae with pan-neuronal expression of Dendra-α-SYN-wt and A53T represented in Appendix A. No changes were found in the number of puncta. Data represented as mean (red line) ± s.d. (*N* = minimum of 6 fish, ns *p* > 0.05) (**C**) Thioflavin-S staining of longitudinal sections through the brain of 3 d.p.f embryos with pan-neuronal expression of Dendra-α-SYN-wt and A53T. Positive staining was observed in α-SYN -wt and A53T fish in contrast to the absence of labelling in transgene-negative siblings or Dendra-tau-A152T expressors. Panels i-iv correspond to magnified images where arrowheads point to thioflavin-s positive structures. Scale bar represents 100 µm. (**D**–**F**) Representative confocal images from maximum projections of z-stacks of Dendra positive neurons in the spinal cord of mosaic Dendra-α-SYN-wt and A53T expressors at 22 h.p.f., showing the early presence of Dendra-α-SYN aggregates (white arrowheads in (**D**)), de novo formation of Dendra-α-SYN-A53T aggregates (red arrowhead in (**E**)), and the increase in size of aggregates within 3 h (31–34 h.p.f.) (yellow arrowheads in (**F**)). Scale bar represents 25 µm.

**Figure 4 genes-13-00868-f004:**
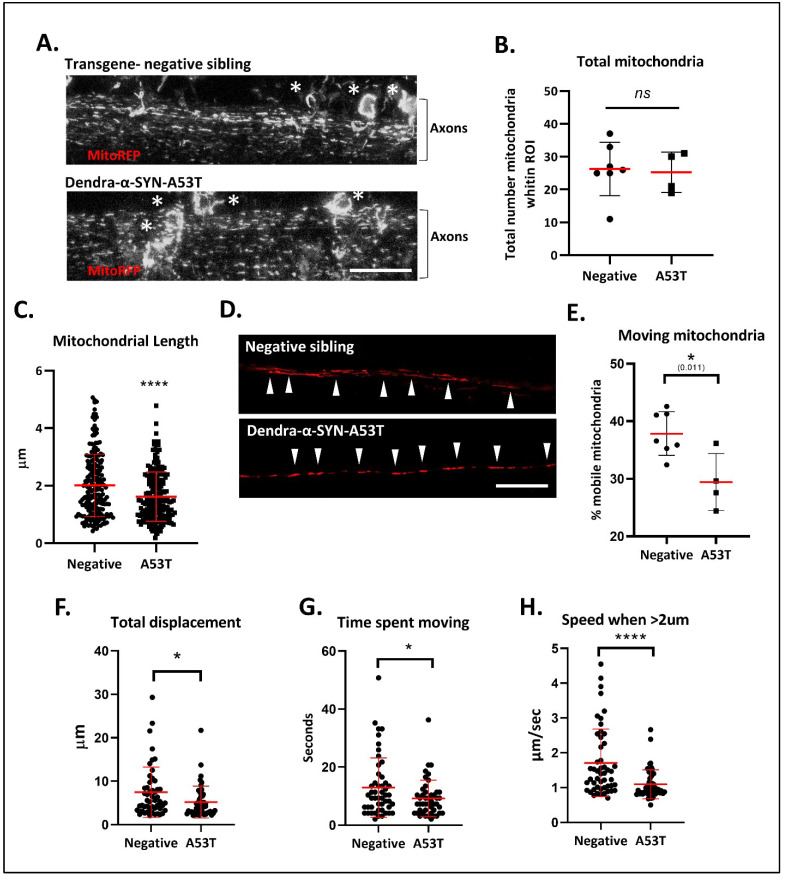
**Defects in mitochondrial morphology and axonal transport.** (**A**) Representative confocal images (max intensity z-stack projections) of labelled mitochondria in the spinal cord 72 h after the injection of *NeuroD*:mitoRFPTag construct into Dendra-α-SYN-A53T with pan-neuronal expression and transgene-negative siblings. Labelled mitochondria could be observed in the soma (white *) and the axonal projections (Axons). Scale bar represents 50 μm. (**B**) Graph represents the total number of mitochondria quantified within the same ROI area, localised to the center of the spinal cord in transgene-negative fish (Negative) and Dendra-α-SYN-A53T fish (A53T). No changes in total number were found, allowing the analysis of mitochondrial morphology and transport in comparative experiments. Data represented as mean (red line) ± s.d. (N ≧ 4 fish; ns *p* > 0.05 vs. negative) (**C**) Quantification of mitochondrial length within the same ROI. Mitochondria in Dendra-α-SYN-A53T fish were significantly smaller than in negative siblings. Data represented as mean (thick red line) ± s.d. (*N* = 200 mitochondria from minimum 4 fish; **** *p* < 0.0001 vs. negative) (**D**) Representative confocal images of RFP-positive mitochondria in a single plane of focus showing individual axonal tracks in transgene-negative and Dendra-α-SYN-A53T fish injected with *NeuroD*:mitoRFPTag construct. Scale bar represents 20 μm. (**E**) Graph representing the percentage of mobile mitochondria in single axons used for the motility analysis in videos. Data represented as mean (red line) ± s.d. (*N* ≧ of 4 videos from independent fish within the same group; * *p* < 0.01 vs. negative). (**F**–**H**) Quantification of mitochondrial motion across 125 μm of spinal cord axon in 1.03 s intervals. Overall mitochondrial transport was significantly reduced in Dendra-α-SYN-A53T-expressing axons compared to transgene-negative siblings. Mobile mitochondria in Dendra-α-SYN-A53T showed significantly less total displacement (**F**), less time moving (**G**) and, consequently, a slower speed (**H**). Data represented as mean (red line) ± s.d. (*N* = 46 mitochondria per group; (* *p* < 0.05, **** *p* < 0.0001 vs. negative).

**Figure 5 genes-13-00868-f005:**
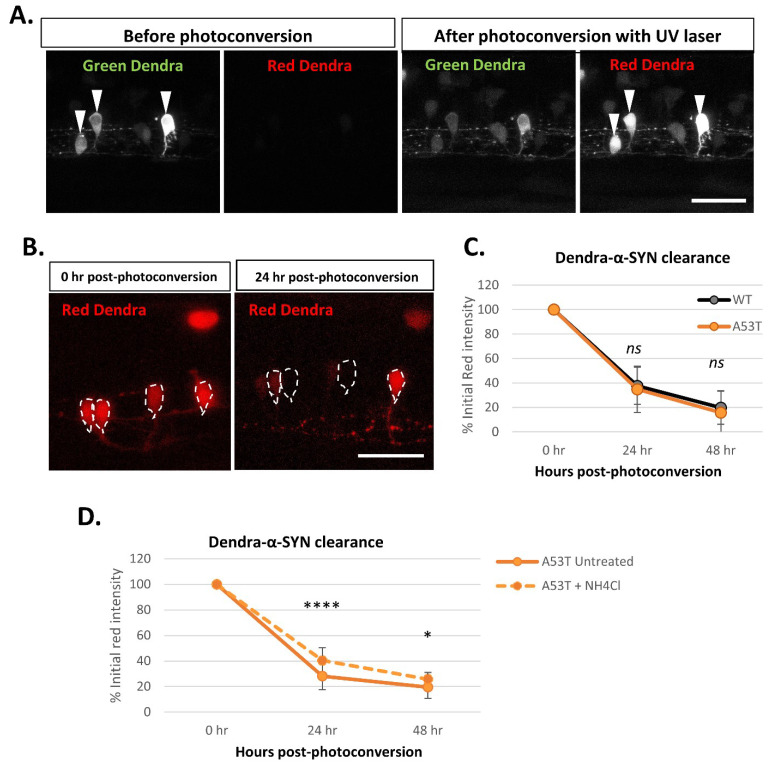
**Dendra-α-SYN clearance kinetics in vivo.** (**A**) Confocal images of the green and red fluorescent signal of Dendra-α-SYN in isolated neurons in the spinal cord of fish with mosaic expression of Dendra-α-SYN-A53T. Neurons with transgene-expression can be identified by its dendra-derived green signal. After UV (405 nm) exposure targeting the soma of selected neurons (white arrowheads), part of Dendra protein is photoconverted from green to red. As a consequence, the pool of green protein is reduced and red Dendra-α-SYN can be visualised. Scale bar represents 50 μm. (**B**) Confocal images showing the reduction in the red signal of photoconverted Dendra-α-SYN-A53T over time. Newly synthesised Dendra-α-SYN protein is green, whereas red signal corresponds to the residual red photoconverted- protein after UV exposure. Red signal within neurons (delimited by dashed-lines) reduces over time as a consequence of its degradation and can be used as a readout for clearance kinetics. Scale bar represents 50 μm. (**C**) Graph representing the decrease in the red-Dendra intensity in single neuronal cells in the spinal cord of Dendra-α-SYN-wt and A53T fish with mosaic expression. Dendra-α-SYN-wt and A53T clear at the same rate (*N* ≧ 30 neurons per group). (**D**) Changes in the clearance rate of Dendra- α-SYN-A53T after inhibition of autophagic flux by ammonium chloride (NH_4_Cl). NH_4_Cl delays the clearance of Dendra-α-SYN-A53T (*N* = minimum 30 neurons per group; * *p* < 0.05, **** *p* < 0.0001 vs. untreated α-SYN-A53T).

## Data Availability

Data is contained within the article or Appendix A provided.

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
