# Peer review of "A New Zebrafish Model to Measure Neuronal α-Synuclein Clearance In Vivo"

_genes, 2022, doi:10.3390/genes13050868_

Round 1

Reviewer 1 Report

The manuscript “A new zebrafish model to measure neuronal α-synuclein clearance in vivo” by Ana Lopez and colleagues aims at presenting a novel and simple model to study a-synuclein aggregation and clearance in neurons. The zebrafish model presented exhibits high expression of Dendra2-A-Synuclein or Dendra2-A53T A-synuclein in neurons and also in other cells, but only neurons seem to develop aggregates.

For this manuscript the zebrafish model is presented mainly to study the involvement of autophagy in the clearance of wild type and A53T a-synuclein. Even though this zebrafish models seem to be very promising to study a-synuclein aggregation progression and also the involvement of other organelles, membranes and protein degradation systems, the manuscript lacks of essential negative controls and the data presentation, especially of what concerns the different models employed, is not completely clear.

Please consider to integrate the results as following in order to improve the quality of the paper.

  • Introduction “Multiple studies have suggested that A53T α-SYN can contribute to the impairment of all clearance pathways previously mentioned and leads to other pathological features, including α-SYN aggregation and mitochondrial defects [12, 18-21]” – Please describe the contribution of the single protein degradation pathways for the clearance of A53T a-synuclein, and add Morgan et al., 2021 reference (DOI: 10.1016/j.celrep.2021.109099).
  • Results 3.2: the toxic effect of wt and A53T a-synuclein was studied at different timepoints (Figure S2A) and then the analysis was focused on 2 dpf embryos (Figure 2B). The count of apoptic cells at 2 dpf in the two figures is very different (e.g. in wt a-synuclein Figure S2A is ~10 and in Figure 2 is ~50). Which is the difference between the two experiments?
  • Results 3.2: the size of the zebrafish is measured in 10 weeks old fishes (Figure 2C) and the mRNA levels of Dendra was measured on 16 weeks old fishes. It would be interesting to see a longitudinal analysis of Dendra-A-synuclein expression in the brain at the timepoints when size ad survival rate is studied.
  • Results 3.2 (Figure S2D): are there any differences in terms of axon length expressed in µm in these fishes?
  • Results 3.3: Figure 3A clearly show the presence of Dendra-a-synuclein-positive aggregates in the axonal projection. To evaluate whether the presence of aggregates is directly related to the overexpression of a-synuclein, fishes expressing Dendra alone should be analyzed.
  • Results 3.3: Figure 3C shows the staining for Thioflavin s in the brain of 3 dpf fishes. As emission spectra of Dendra (peak: 507 nm) and Thioflavin S (peak: 428 nm) partially overlaps, and the images seems to show both Dendra signal and punctuate Thioflavin S signal, I strongly suggest to change this image with a double channel image, in order to be able to discriminate Dendra signal from Thioflavin signal.
  • Results 3.4: in this section the results about the effect of wt a-synuclein expression on mitochondrial transport are not presented. As also this model exhibit aggregation, it would be interesting to see the effect of wt a-synuclein expression on mitochondrial transport.
  • Results 3.5: this section lacks of the comments on Figure 5D that have been probably misplaced in the legend of Figure 5. Please correct this oversight. Moreover, also in this case the analysis of Dendra alone expression should be added to confirm that the decrease of Red Dendra intensity is caused by the clearance of a-synuclein.
  • Results 3.5: Figure 5D clearly shows that autophagy inhibition slows the clearance kinetics of A53T a-synuclein. What about wt a-synuclein? How autophagy inhibition affect the presence of wt a-synuclein in the neurons?
  • Discussion “Other zebrafish models expressing a-syn have been generated previously […]”: please discuss the following reference Weston et al., 2021 (DOI: https://doi.org/10.1016/j.nbd.2021.105291).
  • Discussion “Whether the aggregates seen in these models are LB-like structures would require further validation by structural analysis or the presence of other LB-related proteins in those aggregates”: aggregates present in axons are not referred as LBs, as LB are present in the cell soma, whereas Lewy neurites are usually present in axons. Please correct this oversight.

Reviewer 2 Report

In this manuscript, the authors present the generation of a new zebrafish model of Parkinson’s disease with the aim of to evaluate the degradative capacity of α-synuclein by intact neurons in vivo. They make the fusion of the human SNCA gene for promote the expression of a single fluorescently labelled α-synuclein (Dendra-α-synuclein). This strategy lets the image and quantification of α-synuclein clearance rates in living fish thanks to the photoconversion of the Dendra-α-synuclein, which is a very interesting strategy.

The manuscript is well written and is easy to read. And I have only a number of questions that I think would be interesting:

  • In the abstract, not all the pathological mutations of α-synuclein promote aggregation of the protein; some of them slow down aggregation. In the last case, people believe that the toxicity could be promoted because the amyloid cascade would has a bottleneck that would favour toxic oligomers. This is again mentioned in the beginning of the introduction.

  • The authors introduce the acronym CNS in the introductory section without specifying that it stands for Central Nervous System.

  • On the post-translational modifications that α-synuclein might undergo in the zebrafish model, the authors have investigated phosphorylation at residue 219. They have examined whether the protein is N-acetylated? Around 90% of α-synuclein in humans has this modification. It is just a question that might be interesting to know if they have an answer or why they have not been interested in this modification in their study. I am not suggesting any new experiment or experimental modification.

Overall I find the paper very interesting and the zebrafish model very promising.
